# Probabilistic Traffic Motion Labeling for Multi-Modal Vehicle Route Prediction

**DOI:** 10.3390/s22124498

**Published:** 2022-06-14

**Authors:** Alberto Flores Fernández, Jonas Wurst, Eduardo Sánchez Morales, Michael Botsch, Christian Facchi, Andrés García Higuera

**Affiliations:** 1Fakultät Elektro- und Informationstechnik, Technische Hochschule Ingolstadt, Esplanade 10, 85049 Ingolstadt, Germany; jonas.wurst@thi.de (J.W.); eduardo.sanchezmorales@thi.de (E.S.M.); michael.botsch@thi.de (M.B.); christian.facchi@thi.de (C.F.); 2Escuela Técnica Superior de Ingeniería Industrial, Universidad de Castilla-La Mancha, Calle Altagracia 50, 13001 Ciudad Real, Spain; andres.garcia@uclm.es or; 3European Parliamentary Research Service, Rue Wiertz 60, B-1047 Brussels, Belgium

**Keywords:** PROMOTING, automated driving systems, autonomous vehicles, multi-modal, motion prediction, route prediction, machine learning, real traffic data

## Abstract

The prediction of the motion of traffic participants is a crucial aspect for the research and development of Automated Driving Systems (ADSs). Recent approaches are based on multi-modal motion prediction, which requires the assignment of a probability score to each of the multiple predicted motion hypotheses. However, there is a lack of ground truth for this probability score in the existing datasets. This implies that current Machine Learning (ML) models evaluate the multiple predictions by comparing them with the single real trajectory labeled in the dataset. In this work, a novel data-based method named Probabilistic Traffic Motion Labeling (PROMOTING) is introduced in order to (a) generate probable future routes and (b) estimate their probabilities. PROMOTING is presented with the focus on urban intersections. The generation of probable future routes is (a) based on a real traffic dataset and consists of two steps: first, a clustering of intersections with similar road topology, and second, a clustering of similar routes that are driven in each cluster from the first step. The estimation of the route probabilities is (b) based on a frequentist approach that considers how traffic participants will move in the future given their motion history. PROMOTING is evaluated with the publicly available Lyft database. The results show that PROMOTING is an appropriate approach to estimate the probabilities of the future motion of traffic participants in urban intersections. In this regard, PROMOTING can be used as a labeling approach for the generation of a labeled dataset that provides a probability score for probable future routes. Such a labeled dataset currently does not exist and would be highly valuable for ML approaches with the task of multi-modal motion prediction. The code is made open source.

## 1. Introduction

Urban mobility and transportation are cornerstones of society. Due to the high socio-economic impact of road accidents, there is a motivation to continuously make improvements with regard to automotive safety. This motivation has derived from the development of the modern road infrastructure, which has brought major advances in terms of road safety and traffic-flow efficiency. Recent European Union (EU) road safety statistics [1] show, however, that these improvements stagnated in 2019. Specifically, they quantify a decrease in fatal accidents of 23% when compared to 2010 and of 2% when compared to 2018. For this reason, the EU has launched an ambitious initiative called “Vision Zero” [2], in which it establishes the goal of reducing fatalities caused by traffic accidents to near zero by 2050 and sets the target of halving the number of severe accidents by 2030. To this end, the EU initiative highlights the role that vehicle automation and connectivity play in increasing safety. Given that the majority of accidents (94%) are caused by human error [3], the ADSs under development are mainly focused on improving safety by assisting drivers with the early recognition and avoidance of dangerous situations, while also considering other aspects such as emissions reduction, driving efficiency, and improved passenger comfort. The deployment of automated driving functions in traffic scenarios in open environments is being carried out progressively. The Society of Automotive Engineers defines six levels of automation from levels 0 to 5 [4], where level 5 corresponds to full and unsupervized autonomy. A level 5 automated vehicle demands a very high technological complexity, and, to date, the driving functions required for this level of automation do not have the necessary robustness for deployment in traffic scenarios in open environments. According to [5], the main aspects and systems related to ADSs can be summarized using ten categories: (1) connected systems, (2) end-to-end driving, (3) localisation, (4) perception, (5) assessment and motion prediction, (6) planning, (7) control and dynamic, (8) human machine interface, (9) dataset and software, and (10) implementation. In this work, multidisciplinary research is performed, covering mainly aspects from categories 5 and 9.

A recent line of research [6,7,8,9,10,11,12] focuses on multi-modal motion prediction. This is based on the consideration that traffic motion is multi-modal in nature, meaning that each traffic participant is not bound to follow a single trajectory in the future, but it can instead choose from a wide variety of possible trajectories. In this way, not just one, but multiple probable motion hypotheses are predicted for each traffic participant, allowing researchers to capture the different options a driver may take, such as turning left, making a U-turn or continuing straight ahead, among others. In the following, the term mode refers to a specific estimation of future motion within a finite set of possibilities, and the likelihood that a given mode will be selected is denoted as mode score or mode probability. One prominent approach to address multi-modal motion predictions makes use of ML methods based on the supervised learning paradigm. For this, a labeled dataset is necessary, i.e., the label associated with each sample is known. In case the dataset is generated from real traffic data, only a single real trajectory per traffic participant can be labeled, namely the one that has been driven. This shows the challenge of (a) predicting multiple motion hypothesis for each traffic participant, out of a single labeled one. In addition, the prediction of multiple motion hypotheses implies the assignment of a probability score to each one with respect to the total number of hypotheses. However, labeled datasets with probabilities for routes are not available, (b) resulting in a lack of ground truth for this probability scores.

These aspects ((a) and (b)) motivate the investigation of a method that addresses the following research questions: (1) how to extract the route (certain sections of the road) that represents each possible mode from real traffic datasets, (2) how to estimate the probability that a vehicle will drive a certain mode, and (3) how to generate an adequate multi-modal labeled dataset so that a ML model can learn from it the intrinsic multi-modal motion of traffic scenarios.

In this regard, this work introduces a novel data-based method named PROMOTING that allows the estimation of multiple routes for each traffic participant and provides a probability score for each of the possible future routes. In this way, PROMOTING can be used as a labeling approach for the generation of a labeled dataset that contains not only single trajectories as its ground truth, but also the multiple estimated routes. Given the fact that the early introduction of smart intersections will be of mixed traffic, i.e., automated and non-automated driving together, the modeling of the traffic flow at such scenarios will be significant. The smart intersection is a concept aimed at improving the safety and traffic flow of intersections. It is based on the use of sensors and communication systems that allow researchers to capture and analyze traffic to support ADSs functions. Thus, PROMOTING focuses on urban traffic scenarios, paying special attention to urban intersections.

Therefore, this work makes a contribution to the improvement of multi-modal motion predictions by introducing the PROMOTING method, highlighting the following. First, the method is able to extract multiple motion hypotheses for each traffic participant. Second, the method is able to estimate the probability that a vehicle will drive following a specific motion hypothesis. Third, the method may be used for the generation of a labeled dataset that provides extra information that is useful for a multi-modal prediction task. Fourth, the method is evaluated using real-world traffic scenarios from a database, which allows us to obtain a realistic representation of the traffic’s behavior in urban traffic scenarios.

The rest of the paper is structured as follows: in Section 2, related works are presented. In Section 3, the methodology of PROMOTING is detailed. In Section 4, the evaluation of PROMOTING is presented, and the associated results are shown and described. In Section 5, the main findings of the work are discussed. The paper is summarized in Section 6.

## 2. Related Works

According to [6], the motion prediction of traffic participants can be grouped into the following categories:(1)an engineering approach or physics-based methods,(2)planning-based methods, and(3)pattern-based methods.

Over the last few years, the research into motion prediction has shifted its focus from the physics-based generation of trajectories to the use of ML methods for the same purpose. The authors of [13] proposed the Attention mechanism that marked a shift in the way typical neuro-linguistic programming, time-series forecasting, and sequence-to-sequence problems are approached. Along with this Attention mechanism, Transformer Networks are also finding their way into motion prediction tasks. In [6], Multiple Attention Heads (MAH) are implemented together with a Long–Short Term Memory Encoder–Decoder architecture to predict multiple trajectories, thus addressing the multi-modality of the motion of traffic participants and considering cross-agent interaction modeling. A similar approach is taken in [14]. The difference between [6] and [14] is that the latter adds map-related information that is learned by the Attention mechanism, which assists in modeling the agent–map interaction and improves the system performance. In [15], an architecture based on an Encoder–Decoder structure is proposed, where both are based exclusively on MAH. This model achieves a better performance than the one proposed in [14]. Other recent approaches [9,16,17] build on the work of [13] and use Transformer Networks based on MAH. In [16], pedestrian trajectory prediction is investigated, where the behavior of the pedestrians is modeled without taking into account any kind of interaction with neither traffic participants nor with the map information. This approach is able to closely predict the motion of pedestrians, highlighting the suitability of using Transformer Networks for motion planning tasks. A similar method is presented in [17], where the orientation of the traffic participants is considered to be an additional feature to the input vector when compared to [16]. Furthermore, whereas in [16] only pedestrians are considered, in [17] the performance of the ML model is evaluated for different types of traffic scenarios and different types of road users. A more complex ML-architecture than [16,17] is used in [9], consisting on three stacked Transformer Networks: vehicle motion, vehicle–map interaction, and vehicle–vehicle interaction. The networks are trained sequentially for each epoch, where the vehicle–vehicle interaction network receives the output of the vehicle–map interaction network, and the vehicle–map interaction network receives the output of the vehicle motion network. In addition to receiving the output of the previous one, each network receives additional inputs, which allows each network to specialize in a particular task.

In order for a ML model to learn something as complex as urban traffic, a large amount of data captured from real-world driving scenes is necessary. To prevent over-fitting, the data should have a large variability; in this way, the ML model is able to capture as many as possible of the variations of relevant features.

In the case of urban intersections, for example, the behavior of the traffic participants varies depending on the time of day, working/non-working days, and construction sites, among others. All these situations influence the behavior of the traffic participants, and their consideration provides extra knowledge that must be taken into account by ADSs. On the other hand, capturing real traffic data with these characteristics is a major challenge because of the required financial, computational, and time resources. One strategy to overcome this is to constrain the research and development of ADSs to bounded driving environments, such as smart urban corridors [18]. To this end, it is relevant to use appropriate databases for the training of the ML models.

Current research works [6,7,8,9,10,11,12] that focus on multi-modal motion prediction evaluate their performance either in terms of the Average Displacement Error (ADE), the Final Displacement Error (FDE), or the Root Mean Square Error (RMSE). That is, they consider a single labeled real trajectory and measure the Euclidean distances between the reference trajectory and each of the predicted ones. The best trajectory is then chosen based one of the minimum ADE, the minimum FDE, or the minimum RMSE. The main problem with using these metrics both to reduce training losses and to evaluate the model during the inference phase is that it forces ML models to generate trajectories close to the reference trajectory. This may result in a subset of the predicted trajectories not being drivable, not following the road infrastructure, or colliding with other traffic participants. Furthermore, the prediction of multiple motion scenarios for each traffic participant entails assigning a probability score that indicates the likelihood of selecting a hypothesis within the set of multiple hypotheses; however, the existent datasets containing real traffic data as [19,20,21,22,23,24,25] do not provide this score, as there is only a single real trajectory labeled by each traffic participant.

In [26], the graphs of road topologies are used to identify similar examples through their isomorphism. This is required to shape the latent space for proper novelty detection. Moreover, in [27], the isomorphisms are used to identify similar traffic scenarios, also including the trajectories as paths inside the graphs. As before, this is used for shaping a latent space. However, in the present work, isomorphisms are used to identify similar intersections and routes in the intersections in order to identify similar modes.

Relevant work on the representation of motion hypotheses in traffic scenarios is presented in [28], with the introduction of the Predicted-Occupancy Grids (POGs). These represent the future traffic scenarios in the form of grid cells, where the confidence about the motion of dynamic agents is represented. This approach considers a spectrum of expected occupancy values beyond the simplistic binary approach, i.e., occupied or not occupied. This type of representation is used for the prediction of complex traffic scenarios in [29,30], where different types of machine learning based architectures for POGs estimation are presented. However, there are three notable differences between the work of [28] and the present work. In [28], the approach is based on expert knowledge (assumes physical models of vehicles and motion hypothesis), makes use of simulation data, and the method outputs POGs. In contrast, in the present work, a methodology based on a frequentist approach is proposed (recorded traffic data is analyzed without making a motion hypothesis), real-world traffic data is used, and the presented method (PROMOTING) outputs the modes, in the form of routes, and the mode probabilities.

With regard to all the above, the present research work addresses the shortcomings of multi-modal motion prediction research by proposing the novel PROMOTING method. This serves as the methodology for the generation of a labeled dataset that extracts information about the modes of traffic participants based on conditional prior information. The method is able to extract the number and route of the modes, as well as to estimate the probability that a traffic participant will drive a specific mode. To the best of the authors’ knowledge, this is the first work seeking to estimate the modes with their probabilities in a probabilistic way from real-world data for the purpose of the labeling of multi-modal motion hypothesis.

## 3. Materials and Methods

In order to estimate the modes and the probabilities of each mode, PROMOTING requires (1) historical traffic data and (2) topological information of the road map. To cover these requirements, the publicly available Lyft database [25] is selected, so PROMOTING is evaluated in this work by making use of this database. This database contains traffic motion information that is captured by a vehicle equipped with exteroceptive sensors. It contains a large amount of real-world trajectory data of dynamic participants, including urban intersections, and detailed map information covering the urban area where the traffic scenes were recorded. The methodology of PROMOTING is composed of five steps (see Figure 1), and each step is explained in a subsection of this section.

### 3.1. Road Infrastructure Description

The first step of the PROMOTING method, see Figure 2, aims to describe static traffic information: the road infrastructure. This is described by the road map information contained in the Lyft database on the basis of the map description (see Section 3.1.1) and the intersection description (see Section 3.1.2).

A visual representation of the road infrastructure generated from information from the Lyft database is shown in Figure 3.

#### 3.1.1. Map Description

The road map information contained in the Lyft database divides the road space into so called *ways*, which are road sections of finite length representing an individual lane in a given direction. In this work, each *way* is referred to as a *vertex*. Thus, the set of vertices $\nu_{i}$ of the map *V* is defined as
(1)$$\begin{array}{c} V=\{\nu_{1},\nu_{2},\ldots ,\nu_{i},\ldots ,\nu_{n_{\nu }}\}, \end{array}$$
where *n**_ν_* indicates the order of *G*, i.e., the number of vertices contained in the map. Each vertex $\nu_{i}\in V$ is characterized by a number of features that allow its geometric and connectivity definition, for example:**centreLine:** (x, y) coordinates in global coordinate frame of each vertex.**turnDirection:** indicates the type of change of direction of the vertex: “1” for straight, “2” for left turns, and “3” for right turns.**intersectionId:** unique identifier $\kappa_{i}\in N$ of the intersection of which the vertex forms a part. Value “−1” if the vertex is not part of an intersection.**predecessors:** set $A_{i,\mathrm{pre}}$ that contains the immediate previous vertices with respect to the driving direction, such that $A_{i,\mathrm{pre}}\subseteq V$.**successors:** set $A_{i,\mathrm{suc}}$ that contains the immediate following vertices with respect to the driving direction, such that $A_{i,\mathrm{suc}}\subseteq V$.**leftNeighbours:** set $A_{i,\mathrm{left}}$ that contains the immediate to the left vertices with respect to the driving direction, such that $A_{i,\mathrm{left}}\subseteq V$.**rightNeighbours:** set $A_{i,\mathrm{right}}$ that contains the immediate to the right vertices with respect to the driving direction, such that $A_{i,\mathrm{right}}\subseteq V$.

Thus, the set $A_{i}\subseteq V$ that contains the adjacent vertices of the *i*th vertex is defined as a union of sets, so that
(2)$$\begin{array}{c} A_{i}:=A_{i,\mathrm{pre}}\cup A_{i,\mathrm{suc}}\cup A_{i,\mathrm{left}}\cup A_{i,\mathrm{right}}. \end{array}$$

The connection between the different vertices $\nu_{i}\in V$ provides valuable information for the vehicle motion prediction. In this paper, the connectivity information of the vertices is used to derive a graph-based model that represents the topology of the urban road network. The map topology *G* is then defined as a directed graph, so that
(3)$$\begin{array}{c} G=(V,E), \end{array}$$
where *E* denotes the set of edges $\epsilon_{k}$ of the map, with
(4)$$\begin{array}{c} E=\{\epsilon_{1},\epsilon_{2},\ldots ,\epsilon_{k},\ldots ,\epsilon_{n_{\epsilon }}\}, \end{array}$$
where *n**_ϵ_* indicates the size of *G*, i.e., the number of edges contained in the graph.

Each edge $\epsilon_{k}$ represents the connection between two adjacent vertices, so that
(5)$$\begin{array}{c} \epsilon_{k}=\left\{\left(\nu_{i},\nu_{j}\right)|\;\nu_{i},\nu_{j}\in V\wedge \nu_{j}\in A_{i}\right\},\;i\neq j,\;\forall i,j\in \left\{1,2,\ldots ,n_{\nu }\right\}. \end{array}$$

The order of the vertex pair indicates the driving direction on the edge, where the first element is the “source vertex”, and the second one is the “target vertex”. For example, $\epsilon_{k}=\left(\nu_{i},\nu_{j}\right)\in E$ indicates that the driving direction on the *k*th edge is from the *i*th vertex to the *j*th vertex.

#### 3.1.2. Intersection Description

Similarly, the road topology of an intersection contained in the map, denoted as the ιth intersection, is modeled as the graph $G_{\iota }=\left(V_{\iota },E_{\iota }\right)$ with information from *G* so that $G_{\iota }\subseteq G$. That is, $G_{\iota }$ is a sub-graph of *G*. Therefore, $V_{\iota }\subseteq V$ and $E_{\iota }\subseteq E$.

To model the graph of each intersection, it is necessary to identify which vertices belong to the same intersection and how are they connected to each other. In this sense, three types of vertex are differentiated for each intersection:(1)**Incoming vertex:** The vertex at the entrance of an intersection. These vertices are grouped in sets with the sub-index “in”.(2)**Crossing vertex:** The vertex on an intersection. These vertices are grouped in sets with the sub-index “x”.(3)**Outgoing vertex:** The vertex at the exit of an intersection. These vertices are grouped in sets with the sub-index “out”.

Thus, incoming vertices precede crossing vertices, and crossing vertices precede outgoing vertices. With this, the graph $G_{\iota }$ of the ιth intersection is generated as described in Algorithm 1 and a graphic depiction is shown in Figure 4.


**Algorithm 1:** Intersection graph generation   **Input**: directed graph *G* of the map and the unique intersection identifier ι.   **Output**: directed graph $G_{\iota }$ of the ιth intersection formed by the edges set $E_{\iota }$ and the vertex set $V_{\iota }$ with the incoming, crossing and outgoing vertices of the intersection.**_1_**  
$V_{\iota ,x}:=\left\{\nu_{j}\in V\;|\;\kappa_{j}=\iota \right\}$
**_2_**  
$V_{\iota ,\mathrm{in}}:=\left\{\nu_{i}\in V\;|\;\left(\nu_{i},\nu_{j}\right)\in E\;\forall \;\nu_{j}\in V_{\iota ,x}\right\}$
**_3_**  
$V_{\iota ,\mathrm{out}}:=\left\{\nu_{k}\in V\;|\;\left(\nu_{j},\nu_{k}\right)\in E\;\forall \;\nu_{j}\in V_{\iota ,x}\right\}$
**_4_**  
$V_{\iota }=V_{\iota ,\mathrm{in}}\;⋃\;V_{\iota ,x}\;⋃\;V_{\iota ,\mathrm{out}}$
**_5_**  
$E_{\iota }:=\left\{\left(\nu_{i},\nu_{j}\right)\in E\;|\;\nu_{i},\nu_{j}\in V_{\iota }\;\forall \;i\neq j\right\}$
**_6_**  
$G_{\iota }=\left(V_{\iota },E_{\iota }\right)$



Algorithm 1 can be used for as many intersections as required to generate the set of intersection graphs $S_{\mathrm{int}}$, so that
(6)$$\begin{array}{c} S_{\mathrm{int}}=\left\{G_{1},G_{2},\ldots ,G_{n_{\mathrm{int}}}\right\}, \end{array}$$
where $n_{\mathrm{int}}$ indicates the number of intersection graphs generated from the map.

### 3.2. Vehicle Intersection Data Extraction

Once the intersection graphs $S_{\mathrm{int}}$ and the map vertex set *V* are generated, the next step is the extraction of the list of the Vehicle Intersection Data (VID) $X_{\mathrm{VID}}$. That is, the route information (sequence of vertices) of each vehicle that crosses an intersection and the graph of the crossed intersection. To accomplish this, the motion history of the vehicle is required, in addition to the intersection graphs and the vertex set obtained in the previous step, see Figure 5.

A detailed description of the VID extraction process is depicted in Figure 6.

As depicted in Figure 6, the VID extraction starts by iterating over all $n_{\mathrm{scenes}}$ traffic scenes contained in the Lyft database. Each *i*th scene contains a record of the motion of all $n_{\mathrm{obj}}$ registered objects. For each *i*th scene, the motion information of each *j*th object with “car” label is extracted. Next, for each *j*th vehicle, its (x,y) coordinates are read, and, together with *V*, the coordinates are associated with vertices so as to generate the vertex sequence $Q_{i,j}$, as detailed in Section 3.2.1. Later, $Q_{i,j}$ is used to extract $n_{\mathrm{routes}}$ routes that cross intersections, as detailed in Section 3.2.2. Then, for each $R_{k}$ route that crosses an intersection with graph $G_{\iota }$ a new VID, denoted by $X_{i,j,k}$, is generated. Hence, the VID, $X_{i,j,k}$, for the *k*th route of *j*th vehicle in the *i*th scene is determined as
(7)$$\begin{array}{c} X_{i,j,k}=\left(R_{k},G_{\iota }\right), \end{array}$$
where the route $R_{k}$ is represented by a sequence of vertices and is denoted as follows
(8)$$\begin{array}{c} R_{k}=\left[r_{1},r_{2},\ldots ,r_{q},\ldots \right],\;r_{q}\in V_{\iota }\;\mathrm{and} \end{array}$$
the intersection graph $G_{\iota }$ is generated as indicated by Algorithm 1.

Thus, the VID list $X_{\mathrm{VID}}$ is defined as the list whose elements are the extracted $X_{i,j,k}$ and is denoted as follows
(9)$$\begin{array}{c} X_{\mathrm{VID}}=\left(X_{1},X_{2},\ldots ,X_{n_{\mathrm{VID}}}\right). \end{array}$$

#### 3.2.1. Coordinate–Vertex Association

The first step to extract the $R_{k}$ route is to obtain the vertex sequence $Q_{i,j}$. For this, the (x,y) coordinates of the *j*th vehicle in the *i*th scene at each time instance are associated with vertices contained in *V*. This results in the vertex sequence $Q_{i,j}$, which represents the vertices that the vehicle has driven on. One should note that the association is not unique, meaning that a set of (x,y) coordinates may be associated with multiple vertices, and a vertex may be associated with multiple sets of (x,y) coordinates, which results in a multiple vertex associations. This occurs frequently when the (x,y) coordinates are located at intersections where different crossing vertices overlap. This means that $Q_{i,j}$ must be processed.

First, the invalid (empty) vertex associations are removed from the sequence. An invalid association can happen, for example, when the vehicle moves on “non-drivable” sections of the map. Second, duplicated vertex associations are unified. A duplicated association occurs when a vertex appears in $Q_{i,j}$ in two or more consecutive time instances. By unifying the duplicated vertex associations, only unique ones remain. Finally, $Q_{i,j}$ is filtered according to the intersection topology. This handles multiple vertex associations that can occur when various vertices overlap, see vertices 7, 8, and 9 in Figure 7. Filtering according to the intersection means that only the vertices included and connected in the intersection are kept.

#### 3.2.2. Extraction of Intersection Routes

Once the vertex sequence $Q_{i,j}$ of the *j*th vehicle in the *i*th scene is extracted and processed, the next step is to extract the routes that cross intersections. It is possible for a single vehicle to contain more than one intersection route. An intersection route should fulfill the following two characteristics:1.The route must contain at least one crossing vertex.2.The route must contain either at least one incoming vertex or at least one outgoing vertex.

This approach allows us to differentiate four categories of intersection routes:1.**Complete**: The route contains a full description of how the vehicle approaches, crosses, and leaves the intersection. The route starts with incoming vertices, follows crossing vertices, and ends with outgoing vertices. An example of a complete route is the vertex sequence [2,9,15], see Figure 4.2.**Entering**: The route contains a description of how the vehicle approaches and crosses the intersection. The route starts with incoming vertices and ends with crossing vertices. An example of an entering is the vertex sequence [2,9], see Figure 4.3.**Leaving**: The route contains a description of how the vehicle crosses and leaves the intersection. The route starts with crossing vertices and ends with outgoing vertices. An example of a leaving route is the vertex sequence [9,15], see Figure 4.4.**Other**: Routes that do not belong to any of these three categories. One such route would be that of a vehicle that is standing still during the complete scene, thus remaining at a single vertex. These are omitted, as they do not provide information on how the vehicle approaches or leaves the intersection.

An example of this process is shown in Figure 7. There, the vertex sequence $Q_{i,j}$ of the *j*th vehicle in the *i*th scene is given by
(10)$$\begin{array}{c} Q_{i,j}=\left[4,11,17,22,25,30\right]. \end{array}$$

From $Q_{i,j}$, 11 and 22 are crossing vertices. Intersection IDs κ are taken from these vertices. So, let $\kappa_{11}=8$ indicate that vertex 11 belongs to the 8th intersection and $\kappa_{22}=9$ indicate that vertex 22 belongs to the 9th intersection. Then, the rest of the vertices of the $Q_{i,j}$ vertex sequence that belong to these given intersections are extracted. In this example, the 30th vertex is neglected, as it does not belong to any intersection of this vertex sequence. The remaining vertices are split in as many routes as unique intersection IDs. In this example, two routes are created: one for the 8th intersection and one for the 9th intersection. The elements of the vertex sequence are assigned to a route according to the intersection they belong to. In this example, as the 17th vertex belongs to both the 8th and the 9th intersection, it is assigned to both routes. Next, the type of each vertex of each route is assigned according to the intersection topology. This is the reason why the 17th vertex is assigned as being “outgoing” for the route that corresponds to the 8th intersection and “incoming” for the route that corresponds to the 9th intersection.

### 3.3. Vehicle Intersection Data Clustering

Once the set of intersection graphs $S_{\mathrm{int}}$ and the map vertex set *V* are generated (per Section 3.1) and the VID list $X_{\mathrm{VID}}$ is obtained (as per Section 3.2), the next step is the VID clustering. This step aims to cluster the elements of the list of VIDs $X_{\mathrm{VID}}$ with respect to their graphs. Specifically, the graph isomorphism represents the similarity criterion. Then, the output of this step is $n_{\mathrm{clusters}}$, where each cluster *c* is denoted by $X_{c,\mathrm{VID}}$. A graphical depiction of this process is shown in Figure 8. This process consists of two steps: a pre-clustering of graphs (see Section 3.3.1) and an isomorphic clustering (see Section 3.3.2). These steps are detailed in what follows.

The list $X_{\mathrm{VID}}$ contains $n_{\mathrm{VID}}$ routes of vehicles crossing the intersections; these are classified as they are defined in Section 3.2.2. It should be noted that only complete routes have been selected, because they are the only type of routes that contain a full description of the intersection crossing from the entrance to the exit.

#### 3.3.1. Pre-Clustering

The process of clustering based on isomorphism is computationally expensive. This is specially relevant for large databases, where graph-wise and vertex-wise associations are verified. A brute-force search for the $n_{\nu }!$ possible bijective functions that satisfy the definition of isomorphism between all extracted graphs is not practical.

For this reason, pre-clustering the graphs prior to the isomorphic clustering (Section 3.3.2) is proposed. This is performed by examining a series of preconditions that two graphs must possess in order to be isomorphic. The preconditions are evaluated in a hierarchical manner, allowing us to structure the database in the form of a tree. This database tree allows further analysis of the distribution of the data in terms of graph properties. Then, the first four hierarchical levels of the database tree are detailed in what follows. Alongside this, an example slice of such a database tree is shown in Figure 9.

**Level 0:** The root node of the database tree is located at this level and is the highest hierarchical level from which all branches emerge. All VIDs are inside the root node.**Level 1:** The graphs are grouped by their order, i.e., the number of vertices contained in the graph. Hence, only VIDs with the same graph order are part of the same node. In Figure 9, A and B are two example nodes at that level, with graph orders 20 and 21, respectively.**Level 2:** The graphs are grouped by their size, i.e., the number of edges contained in the graph. VIDs with the same graph orders and sizes are part of the same node. In Figure 9, the node C group VIDs with graph order equal to 20 and graph size equal to 24.**Level 3:** The graphs are grouped by their matrix degree:
(11)$$\begin{array}{c} \Theta_{\mathrm{seq}}=\left[\begin{array}{ccccc} n_{0,\mathrm{in}} & n_{1,\mathrm{in}} & n_{2,\mathrm{in}} & \ldots & n_{n_{\epsilon },\mathrm{in}}\\ n_{0,\mathrm{out}} & n_{1,\mathrm{out}} & n_{2,\mathrm{out}} & \ldots & n_{n_{\epsilon },\mathrm{out}} \end{array}\right], \end{array}$$
where the first row refers to the in-degree of the graph, and the second row refers to the out-degree of the graph, i.e., the number of incoming and outgoing edges to/from the vertices, respectively. With this, $n_{2,\mathrm{in}}$ indicates the number of vertices in the graph whose in-degree is equal to 2, and $n_{2,\mathrm{out}}$ indicates the number of vertices in the graph whose out-degree is equal to 2. Therefore, at this level, only VIDs with the same graph order, the same graph size, and the same matrix degree are grouped. In Figure 9, the node E group VIDs with graph orders equal to 20, graph sizes equal to 24, and matrix degree $\Theta_{1}$.

The levels 0–3 describe the pre-clustering, which creates smaller groups according to their graph properties, such that computationally expensive isomorphism needs to be examined only with the nodes of level 3.

#### 3.3.2. Isomorphic Clustering

Given the database tree from the pre-clustering, the aim is to identify VIDs with similar graphs. Only level 3 need to be taken into consideration, since isomorphism between the graphs is only possible within nodes of level 3.

Two graphs $G_{1}$ and $G_{2}$ are said to be isomorphic if
(12)$$\begin{array}{c} G_{1}≅G_{2}, \end{array}$$
where Equation (Equation 12) holds true if a bijective function $f:V_{G_{1}}\to V_{G_{2}}$ exists, such that
(13)$$\begin{array}{c} \forall v_{i},v_{j}\in V_{G_{1}};\left(v_{i},v_{j}\right)\in E_{G_{1}}\Leftrightarrow \left(f\left(v_{i}\right),f\left(v_{j}\right)\right)\in E_{G_{2}}. \end{array}$$

This means that every vertex and edge of $G_{1}$ has a unique mapping to a vertex and edge of $G_{2}$. All isomorphic graphs are then clustered in level 4 nodes. Nodes H, I, J, and K of Figure 9 are level 4 nodes.

### 3.4. Route-Type Counting

Once the set of intersection graphs $S_{\mathrm{int}}$ and the map vertex set *V* are generated (as per Section 3.1), the VID list $X_{\mathrm{VID}}$ is obtained (see Section 3.2), and the VIDs are clustered (see Section 3.3), the next step is the counting of route types. For this, each *c*th cluster of $X_{\mathrm{VID}}$ is analyzed in order to extract (1) the set of route types ${\hat{R}}_{c}$ and (2) the counting list $\rho_{{\hat{R}}_{c}}$, whose elements indicate how often each route type appears in the cluster. A graphical depiction of this process is shown in Figure 10.

If one considers that the names of the vertices are unique, two intersections cannot be compared by the vertex name alone. Therefore, a common vertex representation per cluster is needed. This common representation is achieved in the form of a template graph that is created for each cluster. The graph of the first $X_{\mathrm{VID}}$ of each cluster is taken as the template of that node. Then, the bijective function (Equations (Equation 12) and (Equation 13)) is used to map the rest of the vertices of the routes within the cluster. A graphical depiction of this process is shown in Figure 11. There, the graph $G^{*}$ is the template graph. The route $R_{G_{1}}$ is mapped to $R_{G^{*}}$ using the bijective function $f:V_{G_{1}}\to V_{G^{*}}$.

Once the vertices of the routes within the cluster are mapped to those of the template graph, the route types are extracted. Each route type is a specific vertex sequence in the cluster. Then, the set of route types ${\hat{R}}_{c}$ is generated for each *c*th cluster as follows:(14)$$\begin{array}{c} {\hat{R}}_{c}=\left\{{\hat{R}}_{c,1},{\hat{R}}_{c,2},\ldots ,{\hat{R}}_{c,n},\ldots \right\}, \end{array}$$
where the first sub-index of the elements of ${\hat{R}}_{c}$ indicates the cluster to which the route type belongs, and the second sub-index is an identifier for the type of route within the cluster.

For each route type ${\hat{R}}_{c,n}$ identified, the frequency $\rho_{{\hat{R}}_{c,n}}$ is computed. This frequency represents how often the route type ${\hat{R}}_{c,n}$ appears in the cluster based on the dataset. This information is relevant for the estimation of the probability that a traffic participant will drive a given route. Then, the counting list of the route types $\rho_{{\hat{R}}_{c}}$ is generated for each *c*th cluster as follows
(15)$$\begin{array}{c} \rho_{{\hat{R}}_{c}}=\left(\rho_{{\hat{R}}_{c,1}},\rho_{{\hat{R}}_{c,2}},\ldots ,\rho_{{\hat{R}}_{c,n}},\ldots \right). \end{array}$$

### 3.5. Mode Estimation

Once the set of intersection graphs $S_{\mathrm{int}}$ and the map vertex set *V* are generated (as per Section 3.1), the VID list $X_{\mathrm{VID}}$ is obtained (see Section 3.2), the VIDs are clustered (see Section 3.3), and the set of route types ${\hat{R}}_{c}$, and the counting list $\rho_{{\hat{R}}_{c}}$ are extracted (as per Section 3.4), the next step is to generate the modes and estimate the mode probability. That is, to create a set of routes that a traffic participant can drive for a given intersection type (cluster) and motion history, and to estimate the probability that a given mode will be driven. Thus, for each *c*th cluster, this process extracts the mode data $M_{c}$. A graphical depiction of this process is shown in Figure 12.

First, a set of sub-routes for each route type is generated in ${\hat{R}}_{c}$ in the *c*-cluster. For example, for the first route type in the *c*-cluster ${\hat{R}}_{c,1}$, the set of sub-routes ${\hat{R}}_{c,1}$ is generated, such that
(16)$$\begin{array}{c} S_{c,1}\subseteq {\hat{R}}_{c,1}. \end{array}$$

Since a route is a vertex sequence, each sub-route $s\subseteq S_{c,1}$ is defined as a coherent sub-sequence of vertices of the corresponding route.

As an example of the creation of the set of sub-routes let the map topology correspond to Figure 4 and the intersection belong to the *c*th cluster. Given the set of route types ${\hat{R}}_{c}$,
(17)$$\begin{array}{c} {\hat{R}}_{c}=\left\{{\hat{R}}_{c,1},{\hat{R}}_{c,2},{\hat{R}}_{c,3}\right\}, \end{array}$$
(18)$$\begin{array}{c} {\hat{R}}_{c,1}=\left[1,7,14\right], \end{array}$$
(19)$$\begin{array}{c} {\hat{R}}_{c,2}=\left[1,7,9,15\right], \end{array}$$
(20)$$\begin{array}{c} {\hat{R}}_{c,3}=\left[2,9,15\right]; \end{array}$$
the corresponding sets of sub-routes $S_{c,1}$, $S_{c,2}$, and $S_{c,3}$ for each route type ${\hat{R}}_{c,1}$, ${\hat{R}}_{c,2}$, and ${\hat{R}}_{c,3}$ can be generated, so that
(21)$$\begin{array}{c} S_{c,1}=\left\{\left[1\right],\left[7\right],\left[14\right],\left[1,7\right],\left[7,14\right],\left[1,7,14\right]\right\}, \end{array}$$
(22)$$\begin{array}{c} S_{c,2}=\left\{\left[1\right],\left[7\right],\left[9\right],\left[15\right],\left[1,7\right],\left[7,9\right],\left[9,15\right],\left[1,7,9\right],\left[7,9,15\right],\left[1,7,9,15\right]\right\}, \end{array}$$
(23)$$\begin{array}{c} S_{c,3}=\left\{\left[2\right],\left[9\right],\left[15\right],\left[2,9\right],\left[9,15\right],\left[2,9,15\right]\right\}. \end{array}$$

Second, the set $S_{c}^{*}$ that contains all unique sub-routes of the *c*th cluster is then defined as
(24)$$S_{c}^{*}=S_{c,1}\cup S_{c,2}\cup S_{c,3}.$$

The mode data $M_{c}$ of the *c*th cluster have as many elements, as the sub-routes *s* are driven in the cluster. This means that, for each sub-route $s\in S_{c}^{*}$, an element of $M_{c}$ is computed. Each element of $M_{c}$ contains (1) the set of modes $\mu_{c,s}$ and (2) the estimated probabilities $P(\mu_{m|c,s})$ of each mode $\mu_{m|c,s}$ and is computed as follows:1.The set of modes $\mu_{c,s}$ used to forecast the possible modes that a vehicle can drive on (1) given the observation of the sub-route *s*, (2) where each mode ends with an outgoing vertex, and (3) where each mode is part of $S_{c}^{*}$. For this, a set ${\hat{S}}_{c,s}$ is created, so that
(25)$$\begin{array}{c} {\hat{S}}_{c,s}=\left\{{\hat{s}}_{1|c,s},{\hat{s}}_{2|c,s},\ldots ,{\hat{s}}_{m|c,s},\ldots \right\}, \end{array}$$
with
(26)$$\begin{array}{c} {\hat{S}}_{c,s}\subseteq S_{c}^{*}:s\in \left\{{\hat{s}}_{1|c,s},{\hat{s}}_{2|c,s},\ldots ,{\hat{s}}_{m|c,s},\ldots \right\}\;,\;\left({\hat{s}}_{m|c,s}\cap V_{c,\mathrm{out}}\right)\neq \emptyset ,\forall m, \end{array}$$
where the set $V_{c,\mathrm{out}}$ contains the outgoing vertices of the template graph of the *c*th cluster. Since the observed sub-route is not part of the modes, i.e., of the future motion, the observed sub-route *s* is extracted from each of the *m*th sub-sequences ${\hat{s}}_{m|c,s}$, generating the corresponding *m*th mode $\mu_{m|c,s}$. This allows the definition of the set of modes $\mu_{c,s}$ as follows
(27)$$\begin{array}{c} \mu_{c,s}=\left\{\mu_{1|c,s},\mu_{2|c,s},\ldots ,\mu_{m|c,s},\ldots \right\}, \end{array}$$
where each element of $\mu_{c,s}$ represents a unique mode of completing the crossing of an intersection with the template graph of the *c*th cluster according to the recorded data and the observation *s*.2.The conditional probability estimation $P(\mu_{m|c,s})$ of the *m*th mode $\mu_{m|c,s}\in \mu_{c,s}$ is estimated. This represents the probability that a traffic participant will drive on the *m*th mode given the *c*th cluster and the *s*th observed sub-route in this cluster. The conditional probability is given by
(28)$$\begin{array}{c} P\left(\mu_{m|c,s}\right)=\frac{\rho_{m|c,s}}{\rho_{c,s}}. \end{array}$$One the one hand, $\rho_{m|c,s}$ indicates how often a vehicle is traveling a route type in the *c*th cluster with the initial sequence-part defined by the *s*th observed sub-route, and the final sequence-part defined by the *m*th mode $\mu_{m|c,s}$. On the other hand, $\rho_{c,s}$ indicates how often a vehicle is traveling the *s*th observed sub-route in the *c*th cluster of the dataset. Then, $\rho_{c,s}$ is defined by
(29)$$\begin{array}{c} \rho_{c,s}=\sum_{q=1}^{|{\hat{R}}_{c}|}\rho_{{\hat{R}}_{c,q}}\cdot z_{q}, \end{array}$$
where
(30)$$\begin{array}{c} z_{q}=\left\{\begin{array}{cc} 1, & s\in S_{c,q}\\ 0, & \mathrm{otherwise}. \end{array}\right. \end{array}$$The frequency $\rho_{{\hat{R}}_{c,q}}$ was introduced in Section 3.4 and indicates how often a vehicle is traveling the route type ${\hat{R}}_{c,q}$ in the *c*th cluster. The Boolean $z_{q}$ allows us to select only those route types ${\hat{R}}_{c,q}$ in which the sub-route *s* is part of its sequence. Given the above, the sum of the probabilities of all modes is then given by
(31)$$\sum_{m=1}^{|\mu_{c,s}|}P\left(\mu_{m|c,s}\right)=1.$$

These two steps (Equations (Equation 25)–(Equation 30)) are applied for each observed sub-route *s* in the *c*th cluster in order to generate each element of the mode data $M_{c}$.

## 4. Evaluation and Results

In this section, the evaluation procedure and evaluation results are detailed. The proposed methodology is evaluated with respect to its ability to generate similar modes, mode probabilities, route types, graphs, and database trees, given similar datasets as inputs. For this, the Lyft database is used as data source, because it contains map information, as well as data about the motion of traffic participants. The data from the traffic participants are randomly divided into two independent datasets ($D_{1}$ and $D_{2}$), where $D_{1}$ is the small training dataset provided Lyft for the Kaggle Challenge https://www.kaggle.com/c/lyft-motion-prediction-autonomous-vehicles (accessed on 11 October 2021), and $D_{2}$ is the validation dataset provided by Lyft, while the map information remains the same for both datasets. An overview of the evaluation process is shown in Figure 13, and each step of the PROMOTING method is detailed in what follows.

The first step of the PROMOTING method (as per Section 3.1) describes the static traffic information (map vertex set *V* and intersection graphs $S_{\mathrm{int}}$). Given that this information does not vary over time and is shared among datasets, the outputs of the first step for each given dataset are not compared. A summary of the road infrastructure description of the Lyft database is shown in Table 1.

The second step of the PROMOTING method (as per Section 3.2) extracts the VID list $X_{\mathrm{VID}}$. Given that each $X_{\mathrm{VID}}$ is generated from a unique set of traffic scenes, the VIDs from different datasets are inherently different. In this step, the routes contained in the VIDs from $D_{1}$ and $D_{2}$ cannot be compared, because the vertices that compose each route have different names and are not yet standardized to a template graph. However, the details of each VID (number of scenes, objects, vehicles, etc.) can be compared, which allows us to corroborate that both $D_{1}$ and $D_{2}$ are similar in size. This is important, because datasets of different sizes would imply different numbers of clusters, types of clusters, modes, and so on. Specifically, a total of ≈1.6 millions routes of vehicles crossing intersections are extracted from the Lyft database [25]. Approximately 50.5% of the routes belong to $D_{1}$, while the remaining ≈49.5% belong to $D_{2}$. The route distribution according to Section 3.2 is shown in Figure 14, and a summary of the details of the VID of each dataset is shown in Table 2.

As can be inferred from Figure 14 and Table 2, both datasets, $D_{1}$ and $D_{2}$, are similar in size, thus aiding in a fair evaluation of the method. Further, as mentioned in Section 3.2, only “complete” routes have been selected in the output of the second step of PROMOTING. The reason for this is that these routes are the only type that contain a full description of the intersection crossing from the entrance to the exit.

The third step of the PROMOTING method (as per Section 3.3) focuses on the clustering of the VIDs according to their graph isomorphism. The comparison metric is the structure of the database trees $T_{D_{1}}$ and $T_{D_{2}}$ that are generated when the datasets $D_{1}$ and $D_{2}$ are used as inputs. The node generation of both trees is analyzed, that is, how was the database tree was generated for each input dataset. If the trees are similar, it is an indication that the method is able to cluster similar routes, even when they come from different datasets. The common tree $T_{\mathrm{com}}$ is defined as one with a lineage such as the one that is present in both $T_{D_{1}}$ and $T_{D_{2}}$, i.e., each node of $T_{\mathrm{com}}$ within each level of the tree has a counterpart in both $T_{D_{1}}$ and $T_{D_{2}}$. $T_{\mathrm{com}}$ can be expressed as follows
(32)$$\begin{array}{c} T_{\mathrm{com}}=T_{D_{1}}\cap T_{D_{2}}. \end{array}$$

The comparison of the structure of the database tree of both $T_{D_{1}}$ and $T_{D_{2}}$ with $T_{\mathrm{com}}$ is shown in Table 3.

Given that both datasets $D_{1}$ and $D_{2}$ are similar in size, from the results shown in Table 3, it can be inferred that the method is able to comparably cluster the dynamic data from different datasets.

The fourth step of the PROMOTING method (as per Section 3.4) consists of the counting of route types within each cluster. Given a cluster *a* from $T_{D_{1}}$, its equivalent cluster *b* from $T_{D_{2}}$ is the one with the similar template graph. The comparison metric is computed by the number of routes in cluster *a* that have an equivalence (same route type) in cluster *b*, normalized by the overall number of routes in cluster *a*. For this, let $n_{c}^{\left(D_{1}\right)}$ be the number of routes of the *c*th cluster of $T_{D_{1}}$ and $n_{c,e}^{\left(D_{1}\right)}$ be the number of similar routes, given the *c*th cluster of $T_{D_{1}}$ and its equivalent cluster in $T_{D_{2}}$. Then, the comparison metric is given by
(33)$$\begin{array}{c} \eta_{c,e}=\frac{n_{c,e}^{\left(D_{1}\right)}}{n_{c}^{\left(D_{1}\right)}}. \end{array}$$

Then, the metric ${\tilde{\eta }}_{c,e}$ that represents the average of the ratio of equivalent routes between all common *c*th clusters from $T_{D_{1}}$ and $T_{D_{2}}$ is estimated as follows
(34)$$\begin{array}{c} {\tilde{\eta }}_{c,e}=\frac{1}{n_{\mathrm{clusters}}}\sum_{c=1}^{n_{\mathrm{clusters}}}\eta_{c,e}. \end{array}$$

For this comparison, ${\tilde{\eta }}_{c,e}=95.82\%$ was achieved. This indicates that common *c*th clusters from $T_{D_{1}}$ and $T_{D_{2}}$ contain mostly the same route types. This indicates that the method is able to cluster the routes of traffic participants from different datasets in a similar manner.

The fifth step of the PROMOTING method (as per Section 3.5) performs the mode estimation. Therefore, the comparison metric is based on the generated modes and their estimated probabilities. For this, let $P(\mu_{m|c,s}^{\left(D_{1}\right)})$ be the probability that a vehicle will drive the *m*th mode given the *c*th cluster and the *s*th observed sub-route, considering the dataset $D_{1}$. Similarly, let $P(\mu_{m_{e}|c_{e},s_{e}}^{\left(D_{2}\right)})$ be the probability that a vehicle will drive the $m_{e}$th mode given the $c_{e}$th cluster and the $s_{e}$th observed sub-route, considering the dataset $D_{2}$. Here, the subscript $\cdots_{e}$ indicates that the corresponding equivalence is used, i.e., the $m_{e}$th mode is the equivalence of the *m*th mode. Therefore, only equivalent modes in equivalent clusters are considered.

Then, the relative difference $\eta_{m|c,s}$ between the probabilities of equivalent modes of both trees with respect to the probability $P(\mu_{m|c,s}^{\left(D_{1}\right)})$ is given by
(35)$$\begin{array}{c} \eta_{m|c,s}=\frac{|P\left(\mu_{m_{e}|c_{e},s_{e}}^{\left(D_{2}\right)}\right)-P\left(\mu_{m|c,s}^{\left(D_{1}\right)}\right)|}{P(\mu_{m|c,s}^{\left(D_{1}\right)})}. \end{array}$$

Equation (Equation 35) is then estimated for all equivalent modes given all equivalent observations in all equivalent clusters. Then, the metric ${\tilde{\eta }}_{m|c,s}$ that represents the average of $\eta_{m|c,s}$ of all equivalent modes for all observations in all common clusters from $T_{D_{1}}$ and $T_{D_{2}}$ is computed as follows: (36)$$\begin{array}{c} {\tilde{\eta }}_{m|c,s}=\frac{1}{n_{m,e}}\sum_{m=1}^{n_{m,e}}\eta_{m|c,s}, \end{array}$$
where $n_{m,e}$ indicates the total number of equivalent modes between $T_{D_{1}}$ and $T_{D_{2}}$. For the used datasets, ${\tilde{\eta }}_{m|c,s}=0.39\%$. This shows that the mode probabilities, when estimated from two different datasets, are similar to each other. This indicates that the mode probability, when calculated using a large dataset, can estimate mode probabilities for similar datasets from same distributions. Even when PROMOTING uses different datasets, it is able to estimate the modes and the probability of each mode in a similar fashion for equivalent sub-route observations in equivalent intersections.

The main results of the evaluation of steps 4 and 5 of PROMOTING are summarized in Table 4.

A representative graphical example of the extraction of modes and the estimation of the mode probabilities is shown in Figure 15.

## 5. Discussion

A common challenge of multi-modal motion prediction, is to determine the “optimal” number of modes to predict, that is, how many trajectories per traffic participant should be predicted in order to comprehensively model a given traffic scene. This question has to take into consideration the amount of computational resources available, the time constraints, and the number of traffic participants, among others. Not only the number of trajectories is important, but also what they should look like. The PROMOTING method serves as a reference that shows both what the modes in a given intersection look like and what the probability is that a traffic participant will drive a specific mode. That is, the proposed method aids in the trajectory-prediction task. The method has the potential to be highly valuable for both the training and inference phases of ML methods for multi-modal motion prediction.

Along with the trajectory prediction that each traffic participant performs, the PROMOTING method could also prove to be useful at smart intersections with Vehicle-to-everything (V2X) capabilities. In that scenario, an automated vehicle could receive the information of the crossing (graphs, modes, etc.) from the infrastructure, so that the traffic participant could perform a better prediction of their own motion according to different parameters, such as efficiency or traffic load. This can be extended to all traffic participants, where each one knows where all the other traffic participants are and can predict the motion of the others with the help of the crossing information. This is relevant in the case of mixed traffic, where automated and human-driven vehicles coexist at the same intersection. Even when no V2X is present, the PROMOTING method could still be on board the EGO-vehicle, and, together with the information from exteroceptive sensors, the relationship between the surrounding traffic participants and their possible routes can be generated.

The PROMOTING method was evaluated in this work using the Lyft database. However, the method is not dependent on this database; instead, it can be used together with other map representations, as long as the required map properties are present, that is, the method is not limited to certain types of intersections but can instead generate the information from many different sources.

The method can be extended using real-time traffic information, as already provided by many navigation tools. The constant update of the traffic conditions (flow, weather, construction works, etc.) can provide an extra benefit for traffic analysis, as well as for the trajectory planning of traffic participants. This real-time traffic information does not necessarily have to come from navigation tools or infrastructure but could also be transmitted by other vehicles in the vicinity that have already crossed the intersection.

It should be noted that the mode probability estimation presented in this work does not take into account the interaction between traffic participants. In this paper, only the past sub-route, not the state of the other objects, are considered in the condition. This is a point for future research, with a special focus on the exchange of intentions between traffic participants via V2X. In addition, the investigation of abnormal behavior of traffic participants is also envisaged.

## 6. Conclusions

In this research work, a novel method named PROMOTING is proposed that is able to generate the modes (probable routes) of traffic participants, as well as estimate the probability that a traffic participant will drive a specific mode. This is done with the aim of supporting ADSs in their task of multi-modal motion prediction.

Mode generation is performed by clustering intersections based on the isomorphisms of their road topology. This allows us to cluster together equivalent intersections and, as a consequence, the equivalent routes of vehicles that crossed the isomorphic intersections. The probability of each mode is estimated based on the frequency with which each route is driven and a given observation (sub-route within the intersection).

The method is evaluated using the Lyft database. The results confirm that the method is able to cluster equivalent intersections and modes. The estimated probabilities of equivalent modes are almost identical, which also corroborates that the method estimates similar probabilities for similar crossings given similar observations. Therefore, PROMOTING provides a methodology that makes it possible to generate a labeled dataset that allows researchers to estimate multiple routes for each traffic participant and provides a probability score for each of the estimated routes. This labeled dataset has the potential to be highly valuable for ML models aimed at the task of motion prediction.

The method could be improved with the inclusion of real-time traffic information that can be sent via V2X communication, including information about the road infrastructure, cellular networks, or other traffic participants. The method is not limited to the used dataset but could also be implemented for other map sources.

Interested readers are referred to the repository [31], where the code that implements the methodology proposed in PROMOTING is made publicly available.

## Figures and Tables

**Figure 1 sensors-22-04498-f001:**
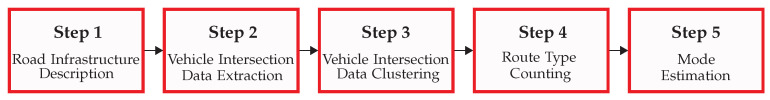
Sequence of operation of PROMOTING method.

**Figure 2 sensors-22-04498-f002:**
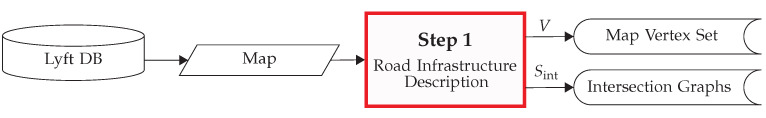
Overview of step 1 of the PROMOTING method: the road infrastructure description.

**Figure 3 sensors-22-04498-f003:**
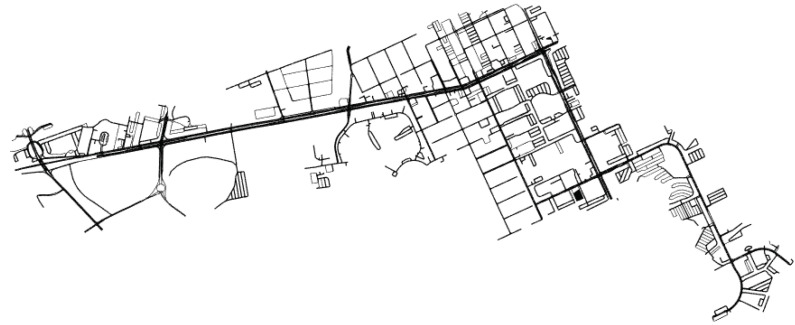
A visual representation of the road infrastructure generated from the Lyft database [25].

**Figure 4 sensors-22-04498-f004:**
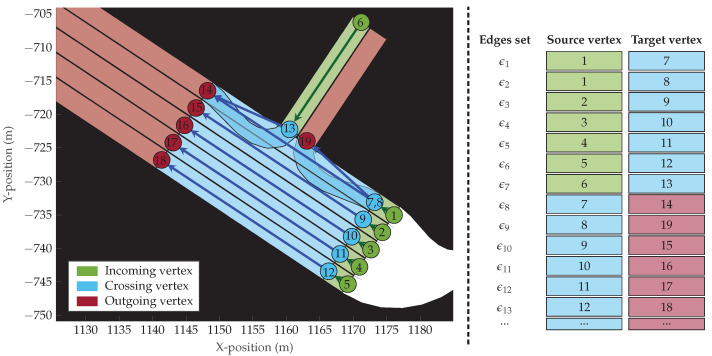
The graphical representation of an intersection. On the (**left**), the vertices are coloured polygons, where the circles represent the first point of the centreLine feature, and the arrows connecting the circles represent the edges. On the (**right**) are the edge matrix with the edge list and the corresponding vertices. In this case, $V_{\iota ,\mathrm{in}}=\left\{1,2,3,4,5,6\right\}$, $V_{\iota ,x}=\left\{7,8,9,10,11,12,13\right\}$, and $V_{\iota ,\mathrm{out}}=\left\{14,15,16,17,18,19\right\}$.

**Figure 5 sensors-22-04498-f005:**
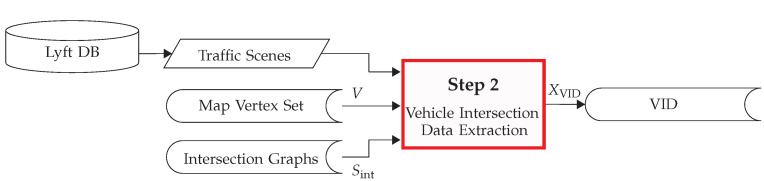
Overview of step 2 of the PROMOTING method: the route extraction.

**Figure 6 sensors-22-04498-f006:**
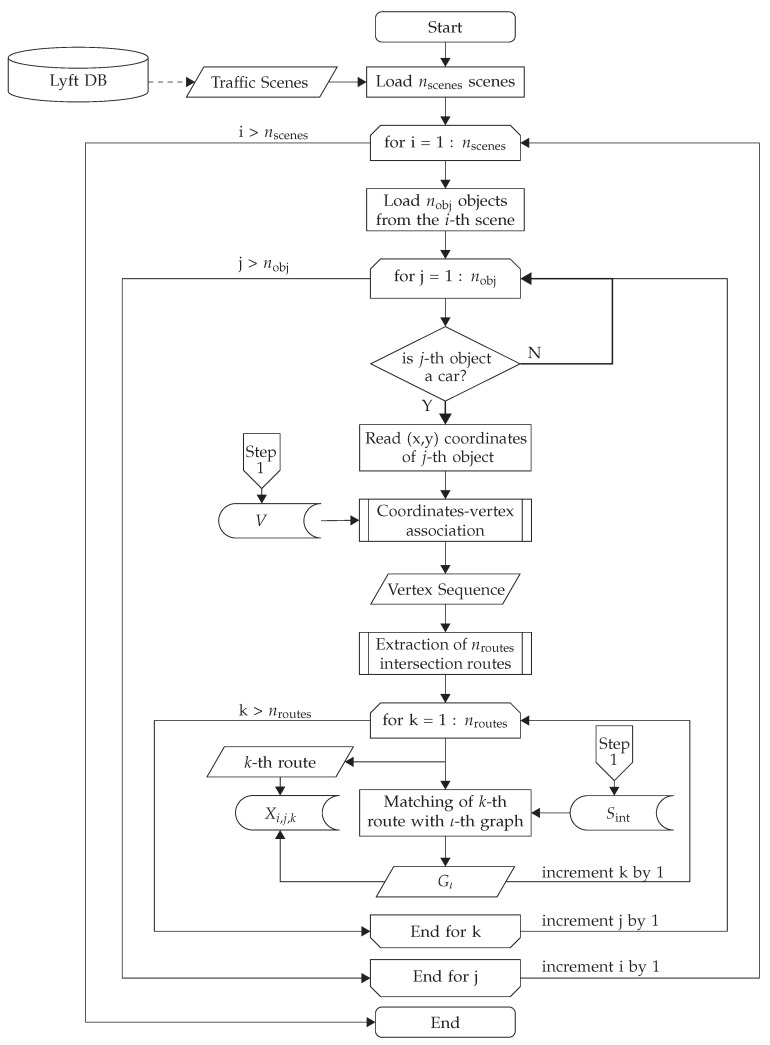
The process to extract the VID.

**Figure 7 sensors-22-04498-f007:**
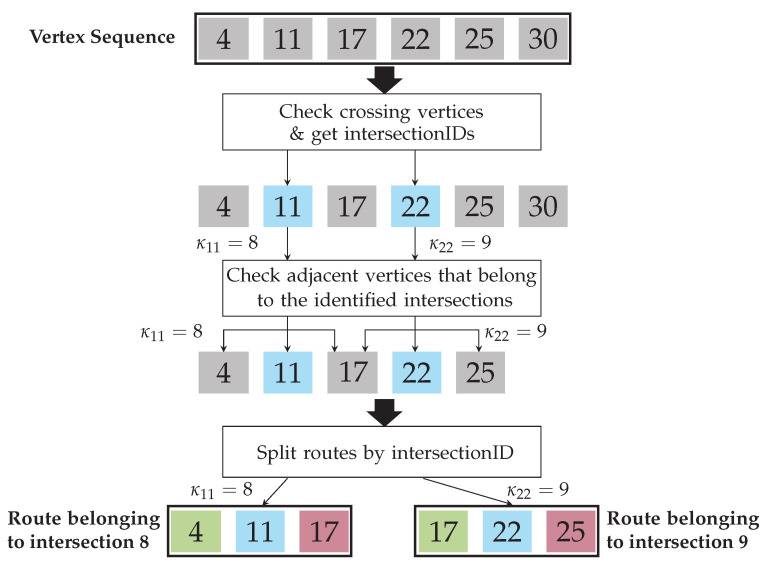
Graphical depiction of the method used to extract intersection routes. Crossing vertices are marked in blue, incoming vertices in green, and outgoing vertices in red.

**Figure 8 sensors-22-04498-f008:**
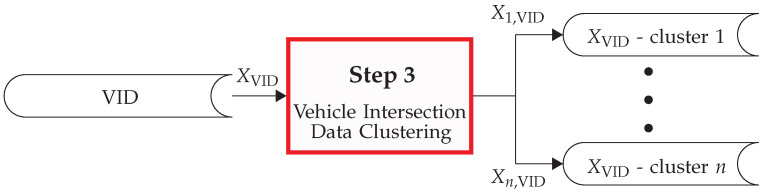
Overview of step 3 of the PROMOTING method: the VID clustering.

**Figure 9 sensors-22-04498-f009:**
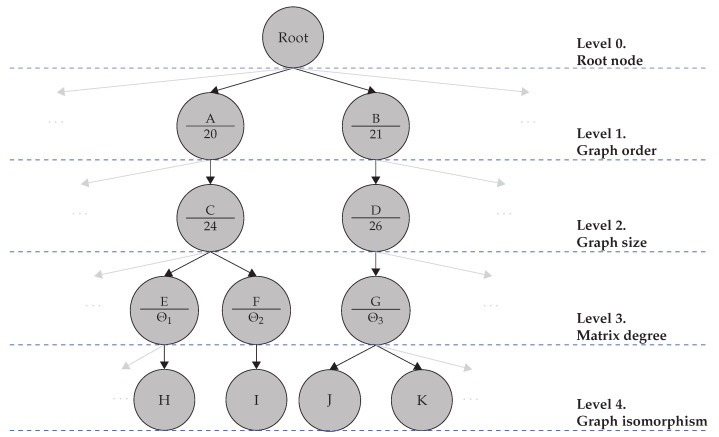
Slice of a database tree for the clustering of the VID based on isomorphic graphs. Each leaf node at level 4 groups the routes of vehicles crossing intersections whose graphs are isomorphic.

**Figure 10 sensors-22-04498-f010:**
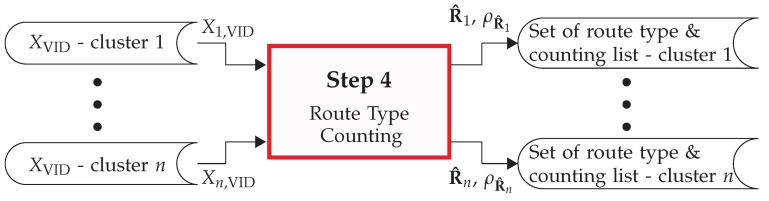
Overview of step 4 of the PROMOTING method: the route-type counting.

**Figure 11 sensors-22-04498-f011:**
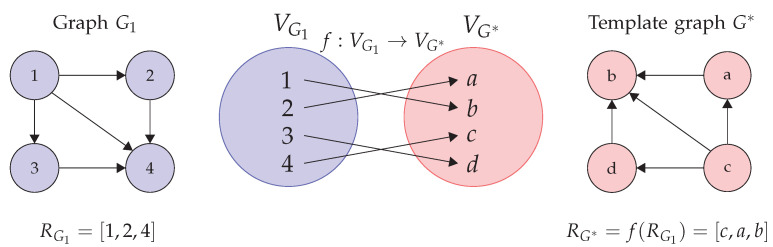
Overview of step 4 of the PROMOTING method: the route-type counting.

**Figure 12 sensors-22-04498-f012:**
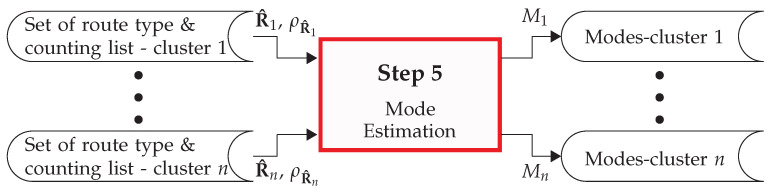
Overview of step 5 of the PROMOTING method: the mode estimation.

**Figure 13 sensors-22-04498-f013:**
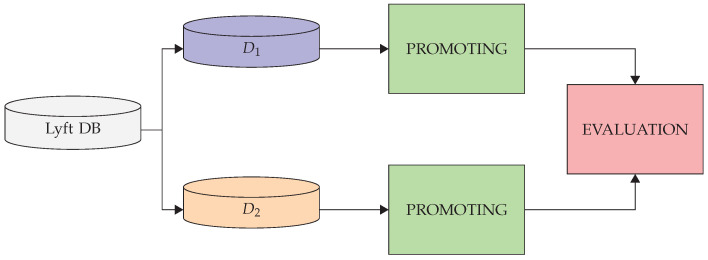
An overview of the evaluation process of the PROMOTING method.

**Figure 14 sensors-22-04498-f014:**
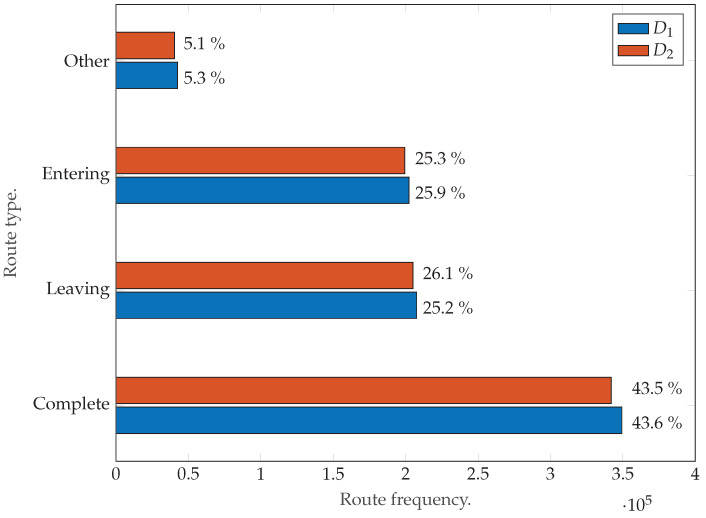
Route distribution according to Section 3.2: complete, outgoing, entering, and other.

**Figure 15 sensors-22-04498-f015:**
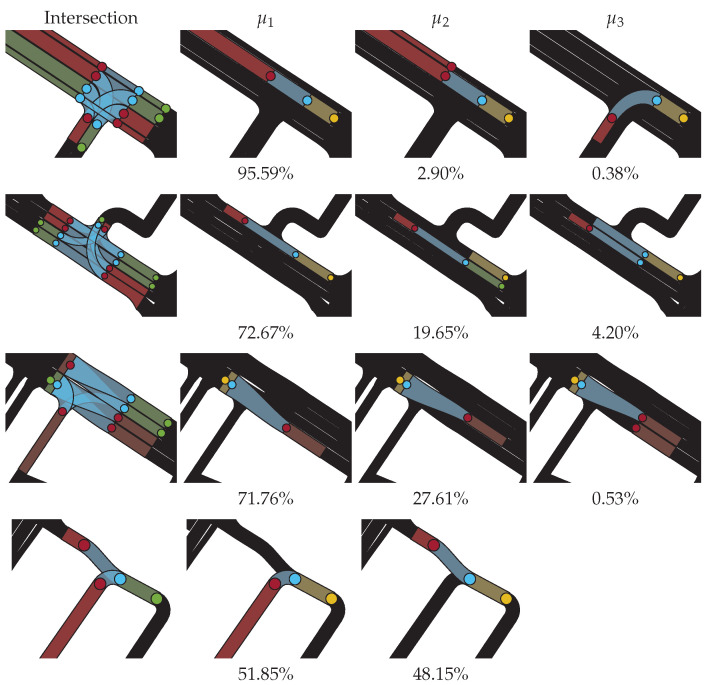
Example of the extraction of modes and estimation of the probability of each mode for four different types of intersections of the Lyft database. In the first column, the intersection is represented by the vertices that compose its graph. The second, third, and fourth columns represent the most probable modes (from highest to lowest probability), given the observed sub-route coloured in yellow and the history of the motion.

**Table 1 sensors-22-04498-t001:** Summary of the road infrastructure description of the Lyft database.

Feature	Name	Value
Graph order (number of map vertices)	*n* _ *ν* _	8506
Graph size (number of map edges)	*n* _ *ϵ* _	12,185
Number of intersections contained in the map	$n_{\mathrm{int}}$	909

**Table 2 sensors-22-04498-t002:** Summary of the details of the VIDs generated from $D_{1}$ and $D_{2}$.

Feature	Name	Value ($D_{1}$/$D_{2}$)
Number of traffic scenes	$n_{\mathrm{scenes}}$	16,265/16,220
Number of traffic participants	$n_{\mathrm{obj},\mathrm{total}}$	20,320,381/19,557,084
Number of vehicles	$n_{\mathrm{veh},\mathrm{total}}$	4,710,949/4,621,107
Number of routes	$n_{\mathrm{VID}}$	801,612/786,919
Number of “complete” routes	$n_{\mathrm{VID},\mathrm{compl}}$	349,322/342,003
Number of “entering” routes	$n_{\mathrm{VID},\mathrm{enter}}$	202,260/199,359
Number of “leaving” routes	$n_{\mathrm{VID},\mathrm{leave}}$	207,595/205,122
Number of “other” routes	$n_{\mathrm{VID},\mathrm{other}}$	42,435/40,435
Number of intersections crossed by “complete” routes	$n_{\mathrm{int},\mathrm{veh}}$	250/250

**Table 3 sensors-22-04498-t003:** Comparison of the database trees of both $T_{D_{1}}$ and $T_{D_{2}}$ with $T_{\mathrm{com}}$.

	Database Tree			
	$T_{D_{1}}$		$T_{D_{2}}$	
**Feature**	$\cap T_{\mathrm{com}}$	**Total**	$\cap T_{\mathrm{com}}$	**Total**
Clusters ($n_{\mathrm{clusters}}$)	168 (97.1%)	173	168 (97.1%)	173
Routes ($n_{\mathrm{VID},\mathrm{compl}}$)	349,313 (99.99%)	349,322	341,997 (99.99%)	342,003

**Table 4 sensors-22-04498-t004:** Main results of the evaluation of steps 4 and 5 of PROMOTING.

Feature	Name	Value
Average ratio of equivalent routes	${\tilde{\eta }}_{c,e}$	95.82%
Average relative difference between equivalent modes	${\tilde{\eta }}_{m|c,s}$	0.39%

## Data Availability

The code is public under [31].

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
