# Peer review of "Probabilistic Traffic Motion Labeling for Multi-Modal Vehicle Route Prediction"

_sensors, 2022, doi:10.3390/s22124498_

Round 1

Reviewer 1 Report

In this research work, authors proposed a novel method named PROMOTING that is able to generate the modes (probable routes) of traffic participants, as well as to estimate the probability that a traffic participant will drive a specific mode. This with the aim of supporting ADSs in their task of multimodal motion prediction. I have the following comments to improve the paper.

1.       Acronyms used for the first time in the manuscript need to be explained such as ML which stands for machine learning used in the abstract on line#5.

2.       Some capital letters are used in the middle of the words. For example, MOtion in line#7, PRObabilistic etc.

3.       A table of notations with an explanation will improve the readability of the paper.

4.       Figures numbers should be mentioned in sequential order. E.g. figure 7 is mentioned before figure 7 in the text. Moreover, figure 15 is referred to first then figure  1.

5.       I am curious why the research has not been compared with previous works?

6.       Mention some more recent works, if possible, from the last 2~3 years.

7.       Improve the presentation of the figures with more details

Reviewer 2 Report

Paper is devoted to a novel method of probabilistic motion labelling for the future motion of traffic participants in urban intersections. It is a very consistent approach, but I have the feeling that some narrative parts could be shortened.  Paper is well documented, I mean the author are mentioning a lot of contributions in Introductions, but they not provide any reference or comparison of their obtained results to other approaches, if they exists. Who else used datasets from Lyft database in similar ways ?
